# Evaluation of a connectivity-based imaging metric that reflects functional decline in Multiple Sclerosis

**Katherine A. Koenig** [1]*, **Erik B. Beall**[1], **Ken E. Sakaie**[1], **Daniel Ontaneda**[2], **Lael Stone**[2], **Stephen M. Rao**[3], **Kunio Nakamura** [4], **Stephen E. Jones**[1], **Mark J. Lowe**[1]

**1** Imaging Sciences, Imaging Institute, Cleveland Clinic, Cleveland, OH, United States of America, **2** Mellen Center, Neurologic Institute, Cleveland Clinic, Cleveland, OH, United States of America, **3** Schey Center for Cognitive Neuroimaging, Neurologic Institute, Cleveland Clinic, Cleveland, OH, United States of America, **4** Lerner Research Institute, Cleveland Clinic, Cleveland, OH, United States of America

* koenigk@ccf.org

**Data Availability Statement:** The data underlying the results presented in this study are available from the figshare database (doi.org/10.6084/m9. figshare.14618265).

## Abstract

Cognitive impairment is a common symptom in individuals with Multiple Sclerosis (MS), but meaningful, reliable biomarkers relating to cognitive decline have been elusive, making evaluation of the impact of therapeutics on cognitive function difficult. Here, we combine pathway-based MRI measures of structural and functional connectivity to construct a metric of functional decline in MS. The Structural and Functional Connectivity Index (SFCI) is proposed as a simple, z-scored metric of structural and functional connectivity, where changes in the metric have a simple statistical interpretation and may be suitable for use in clinical trials. Using data collected at six time points from a 2-year longitudinal study of 20 participants with MS and 9 age- and sex-matched healthy controls, we probe two common symptomatic domains, motor and cognitive function, by measuring structural and functional connectivity in the transcallosal motor pathway and posterior cingulum bundle. The SFCI is significantly lower in participants with MS compared to controls ($p = 0.009$) and shows a significant decrease over time in MS ($p = 0.012$). The change in SFCI over two years performed favorably compared to measures of brain parenchymal fraction and lesion volume, relating to follow-up measures of processing speed (r = 0.60, $p = 0.005$), verbal fluency (r = 0.57, $p = 0.009$), and score on the Multiple Sclerosis Functional Composite (r = 0.67, $p = 0.003$). These initial results show that the SFCI is a suitable metric for longitudinal evaluation of functional decline in MS.

## Introduction

Physical disability and upper extremity function are hallmark symptoms of Multiple Sclerosis (MS), and are among the most important factors affecting quality of life [1, 2]. Another common symptom is cognitive dysfunction, affecting approximately 40% to 65% of patients and impacting employment, daily living skills, and overall quality of life [3–6]. Treatment efficacy is often judged by impact on progression of motor disability [7], with less focus on cognitive

**Funding:** This work was supported by grant number RG4931A1, awarded to M.J.L., from the National Multiple Sclerosis Society (nationalmssociety.org). The funders had no role in study design, data collection and analysis, decision to publish, or preparation of the manuscript.

**Competing interests:** The authors have declared that no competing interests exist.

function. This is partially due to high variability of cognitive measures and to the relatively weak relationship between cognitive measures and commonly used imaging metrics in MS, such as lesion burden [8]. Here, we introduce and evaluate a composite imaging metric for the assessment of both motor and cognitive function in MS, with the ultimate goal of providing a widely-applicable metric suitable for use in longitudinal studies assessing therapeutic impact.

The two principal measures used to probe the symptomatic domains of MS are the Expanded Disability Status Scale (EDSS) and the Multiple Sclerosis Functional Composite (MSFC). Currently, the EDSS is the primary clinical test accepted by clinical trial monitors, despite the fact that the EDSS does a poor job of assessing upper limb function and cognitive deficits [9]. The MSFC probes three symptomatic domains: lower extremity function (ambulation), upper extremity function, and cognition. The cognitive component of the MSFC is based on the Paced Serial Addition Test (PASAT) [10], or, more recently, the Symbol Digit Modalities Test (SDMT) [11]. Here, we introduce an imaging-based metric that is similarly designed to provide a composite measure of neurologic deficit. This metric was develop based on previous work, with the goal of assessing both disease status and functional decline over time. The data presented here represents the first test of this metric in real-world data.

Our previous work focused on the relationship of imaging measures to motor and cognitive symptoms in MS. We reported reduced functional connectivity, assessed by resting state functional magnetic resonance imaging (rsfMRI), between the bilateral primary motor cortices in patients with MS as compared to controls [12]. A follow-up study focused on both structural and functional connectivity, showing that diffusion tensor imaging (DTI) measures in the transcallosal motor pathway were inversely correlated with rsfMRI of the primary sensorimotor cortices (SMC) in MS [13]. Given the frequency of memory impairment in MS [3, 14], we subsequently extended our study of DTI- and rsfMRI-based connectivity to the posterior cingulum bundle, a pathway connecting the antero-mesial temporal lobe (AMTL) and the posterior cingulate cortex (PCC) and subserving episodic memory [15]. In this pathway, as well as within the transcallosal motor pathway, we found an inverse correlation between radial diffusivity (RD) and rsfMRI in both MS and healthy controls [15]. Because RD is generally accepted to reflect the degree of demyelination and axonal loss in MS [16], this relationship suggests that reduced functional connectivity in monosynaptic pathways in the brain is related to the structural integrity of the white matter along that pathway. Despite this, the correlation between functional and structural connectivity in these studies was moderate (r~0.4). This suggests that, although related, the measures reflect different aspects of the biological processes that affect connectivity, and thus may be complementary.

This work presents the results of a two-year study designed to test the hypothesis that a metric based on connectivity measures along network pathways that are implicated in domains of disability specific to MS can be used as a sensitive marker of disease status and functional decline. We focus on two common domains of disability in MS, motor and cognitive function, and characterize them using measures of both structural (DTI) and functional connectivity (rsfMRI). Our previous work has shown significantly reduced structural and functional connectivity of the transcallosal motor pathway in MS [12, 17], as well as a significant correlation between structural and functional connectivity in this pathway [13, 15]. Although these were cross-sectional studies, they informed our choice for the motor disability domain pathway. We hypothesize that measures of transcallosal motor pathway connectivity will be related to measures of motor function over time, and that this relationship will be reflected in our connectivity metric. Based on work relating the posterior cingulum bundle to memory dysfunction and speed of processing in MS [18, 19], we selected the posterior cingulum bundle as the cognitive pathway. The PCC and AMTL have been implicated in episodic memory by multiple investigators [20, 21], and these regions have been shown to be abnormal in memory-impaired

patients with Alzheimer's disease [22–24]. The direct anatomic connection between these regions has been demonstrated in multiple animal studies [25–28], and by using DTI and functional connectivity measurements in healthy human controls [15]. We hypothesize that measures of PCC-AMTL pathway connectivity will be related to measures of cognitive function over time, and that this relationship will be reflected in our connectivity metric.

## Theory

Here, we present an evaluation of this metric, called the Structural and Functional Connectivity Index (SFCI). The SFCI was initially developed using a generated dataset with parameters based on the results of previous work (see S1 Text), and was constructed to measure combined structural and functional connectivity in two commonly-impacted domains of MS–motor and cognitive function. The metric can be normed to either control or patient samples and will decrease with increased connectivity. This provides important flexibility to the metric, as the observed variability of the component measures were higher in MS than in controls. In the current analysis, mean and standard deviations taken from the control group were used to norm Eqs 1 and 2 below. When used in this manner, the SFCI is an evaluation of pathway connectivity in individual MS patients as compared to normative values for healthy controls.

**Motor SFCI metric.** Using measures from the transcallosal motor pathway, the motor SFCI component is calculated according to the following equation:

$$Z_{motor} = \frac{1}{2}\left[\frac{(fc - fc^{pop})}{\sigma_{fc}^{pop}} + \frac{(sc - sc^{pop})}{\sigma_{sc}^{pop}}\right] \tag{1}$$

where $fc$ and $sc$ are functional and structural connectivity, respectively, for an individual, $fc^{pop}$ and $sc^{pop}$ are the mean functional and structural connectivity, respectively, of the normative sample, and $\sigma_{fc}^{pop}$ and $\sigma_{sc}^{pop}$ are the standard deviations of functional and structural connectivity, respectively, of the normative sample.

**Cognitive SFCI metric.** Because the PCC-AMTL pathway is intrahemispheric and bilateral, the cognitive SFCI component ($Z_{cog}$) was constructed to be sensitive to impairment in either hemisphere. The left and right hemisphere PCC-AMTL measures are z-scored separately according to the following equation:

$$Z_{cog}^{L,R} = \frac{1}{2}\left[\frac{(fc - fc^{pop})^{L,R}}{L, R_{\sigma_{fc}^{pop}}} + \frac{(sc - sc^{pop})^{L,R}}{L, R_{\sigma_{sc}^{pop}}}\right] \tag{2}$$

The notation of Eq 2 is identical to that of Eq 1. L = left, R = right. The results are normalized scores for the left ($Z_{cog}^{L}$) and right ($Z_{cog}^{R}$) hemispheres.

**SFCI combined metric.** The motor ($Z_{motor}$) and cognitive ($Z_{cog}^{L}$ and $Z_{cog}^{R}$) SFCI metrics are combined in the following manner:

$$SFCI = \frac{1}{2}Z_{motor} + \frac{1}{2}min\left[Z_{cog}^{R}, Z_{cog}^{L}\right] \tag{3}$$

where $Z_{cog}$ is taken to be the minimum Z score of the two hemispheres.

## Materials and methods

### Data acquisition

**Participants.** Twenty-five participants with MS and 12 healthy controls were enrolled for longitudinal study. Of this sample, twenty participants with MS and 9 healthy controls were

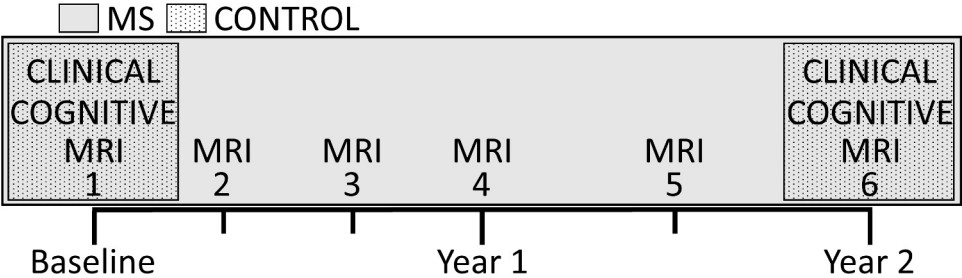

**Fig 1. Schematic of study visits.** MS = Multiple Sclerosis.

included in the final data analysis (see Results: Sample Description for details). Participants with MS completed six study visits over a two year period. Year 1 included four visits spaced at four month intervals, and year 2 included two visits spaced at two month intervals. Controls were selected to provide an age-, education-, and sex-matched sample and completed two study visits—one at baseline and the other after two years. Fig 1 shows the visit timeline and testing schedule. All data were acquired after providing written informed consent under protocol 13–994, approved by the Cleveland Clinic Institutional Review Board (Federalwide Assurance number 00005367). Anonymized data that support the findings of this study is available at: doi.org/10.6084/m9.figshare.14618265.

**Clinical and cognitive evaluation.** The EDSS was measured by an experienced MS neurologist (LS, DO). All participants completed a cognitive battery at baseline and two years, administered by an experienced psychometrist under the supervision of a licensed clinical psychologist (SR). Key tests from the Minimal Assessment of Cognitive Function in MS (MAC-FIMS) were administered, and raw scores for each measure were corrected using published norms (Table 1). To lessen test-retest effects, alternate test forms were counterbalanced in presentation.

**MRI acquisition.** Data were acquired on a Siemens TIM Trio 3T MRI scanner (Erlangen, Germany) using a 12-channel receive-only head array or a Siemens Prisma 3T MRI scanner using a 20 channel head-neck array. Approximately one quarter of the way into data collection, the Trio scanner was upgraded to the Prisma platform. Statistical analyses described below accounted for possible systematic effects of scanner change. A bite bar was used to minimize participant motion. The bite bar consisted of a molded dental impression (Kerr Dental, Inc., Brea CA) taken of the subject's teeth and affixed to a removable plastic frame placed over the head coil.

At baseline, participants performed a unilateral patterned finger tapping task during scanning. One scan was performed for each hand, with left/right order randomized across

**Table 1. Neuropsychological measures.**

| Cognitive domain | Test | Resulting measures |
|---|---|---|
| Verbal episodic memory | California Verbal Learning Test–II (CVLT) | T-score [29] |
| Visuospatial episodic memory | Brief Visuospatial Memory Test–Revised (BVMT) | T-score; delayed recall T-score [30] |
| Processing speed | Symbol Digit Modalities Test (SDMT) | corrected score [31] |
| Executive function | Delis–Kaplan executive function system (D-KEFS) | composite scaled score [32] |
| Spatial processing | Judgment of Line Orientation Test (JLO) | corrected score [33] |
| Expressive Language | Controlled Oral Word Association Test (COWAT) | corrected score [34] |

participants. Participants were asked to tap their fingers as fast as possible while maintaining the following pattern: thumb, middle, pinkie, index, ring. During scanning, fiber optic gloves were used to monitor task performance and the verbal commands "start" and "stop" were used to indicate tap/rest periods.

Directly prior to scanning, participants met with the experimenter. The experimenter described the scanning session, including familiarizing the participant with the bite bar apparatus, highlighting the importance of remaining still during scanning, and describing the finger tapping task. Participants practiced tapping both with and without the fiber optic gloves until they felt comfortable with the pattern.

The following scans were performed in the order shown:

Scan 1: Whole brain (MPRAGE): 176 axial slices at 0.94 mm thickness; field-of-view (FOV) 240 mm×240 mm; matrix 256×256; voxel size 0.94 mm$^3$; inversion time (TI)/echo time (TE)/repetition time (TR)/flip angle (FA) = 1100 ms/2.84 ms/1860 ms/10˚; bandwidth (BW) 180 Hz/pixel. Acquisition time 4:20 minutes. In addition to providing anatomic detail, this scan was used to provide gray and whiter matter masks using Freesurfer segmentation [35].

Scan 2: SPACE 3D FLAIR: 144 sagittal slices at 1.2 mm thickness; FOV 256 mm×224 mm; matrix 256×224; voxel size 1.2×1×1 mm$^3$; TI/TE/TR/FA = 2000 ms/395 ms/6500 ms/120˚; BW 698 Hz/pixel; 6/8 partial Fourier acquisition; GRAPPA factor = 2; 24 reference lines. Acquisition time 5:14 minutes.

Scan 3: Whole brain BOLD resting state scan: 31 axial slices at 4 mm thickness; FOV 256 mm×256mm; matrix 128×128; voxel size 2×2×4mm$^3$; TE/TR = 29 ms/2800 ms; BW 1954 Hz/pixel; 132 repetitions. Data was acquired using a prospective motion-controlled, gradient recalled echo, echoplanar acquisition [36]. Acquisition time 6:10 minutes. Prior to the start of the scan, the participant was instructed to rest with eyes closed and refrain from any voluntary motion.

Scan 4: Whole brain BOLD tapping activation scan (baseline only): Data were acquired using the same parameters as for scan 3, with 160 repetitions. Acquisition time 7:29 minutes, 2 runs. Participants performed a unilateral patterned finger tapping task during four blocks of 16 volume acquisitions (45 seconds), interleaved with rest blocks of the same duration. Prior to the start of the scan, participants were reminded of the tapping pattern and verbal commands used during the task.

Scan 5: High angular resolution diffusion imaging (HARDI): 51 axial slices at 2 mm thickness; FOV 256 mm×256 mm; matrix 128×128; voxel size 2mm$^3$; TE/TR = 92 ms/7800 ms; BW 1628 Hz/pixel; 5/8 partial Fourier acquisition; 71 noncollinear diffusion-weighting gradients with b = 1000 s/mm$^2$, eight b = 0 volumes, NEX = 4. Twice-refocused spin echo was used to minimize eddy current artifact [37]. Acquisition time 10:24, two averages.

## Data analysis

**MRI data processing.** For the fMRI tapping analysis, motion correction was performed using volumetric and slice-based estimators from SLOMOCO [38]. Using in-house code implemented in Matlab R2018b, a 4mm 2D in-plane Gaussian filter was used to improve functional contrast-to-noise ratio and make in-plane and through-plane resolution more similar [39]. The AFNI routine 3dDeconvolve was used to fit a boxcar reference function representing the off/on activation paradigm to the time series data of each voxel [40]. The result was a

whole brain Student's t map, used to determine regions of significant involvement in the uni-manual tapping task.

For the rsfMRI analysis, physiologic signals were regressed using RETROICOR as provided by AFNI [40, 41]. For the majority of studies, a plethysmograph and respiratory bellows were used during scanning to sample cardiac and respiratory signals at 400 Hz. For 15 scans, technical problems prevented acquisition of one or more physiologic signals. In those cases, physiologic signals were estimated with PESTICA [42]. Simultaneously with regression of physiologic noise, data were retrospectively motion-corrected using volumetric and slice-based estimators from SLOMOCO [38]. Mean (TDzmean) and maximum (TDzmax) voxel-level residual displacement were used to characterize motion artifact in each scan for use in summary statistics and quality control. If a participant had motion values of TDzmax > 1mm and TDzmean > 0.2mm at baseline, they would be administratively withdrawn from the study [43]. Follow-up scans over the same threshold were flagged for visual inspection of the rsfMRI time series and resulting correlation maps, detailed below. Using in-house code implemented in Matlab R2018b, spatial filtering with a 2D in-plane Hamming filter was performed to improve functional contrast-to-noise ratio with minimal loss of spatial resolution [44] and the data were temporally filtered to remove all fluctuations above 0.08Hz [45, 46].

For the DTI analysis, image series were concatenated, followed by iterative motion correction [47] that included updating of diffusion gradient directions [48]. The diffusion tensor and diffusivities were calculated on a voxel-by-voxel basis. All analyses were completed using in-house software.

**Other MRI measures.** In order to compare to MRI measures that are typically used as outcomes in clinical trials of MS therapies, the MPRAGE and FLAIR were used to measure the following in all participants with MS:

At each visit, lesion volume (LV) was calculated using FLAIR images [49]. A detailed description of lesion volume calculation is available in reference [49]. Briefly, optimal thresholding and connected-components analysis were used to generate a starting point for segmentation. A 3D radial search was then performed to locate probable locations of the intra-cranial cavity. The result was total LV for each MS participant at each visit.

At each visit, brain parenchymal fraction (BPF) was calculated using anatomical images. BPF was defined as the ratio of brain parenchymal volume to the total volume within the brain surface contour. The method used here [50] used the segmentation algorithm described above [49] for brain surface detection and brain volume calculation.

**Seed region definition.** Seeds were defined separately for each subject using baseline data. The AFNI routine align_epi_anat.py was used to perform affine 12 parameter alignment of task-based fMRI and anatomical volumes to rsfMRI and DTI volumes [51]. The same routine was used to align baseline images to the same imaging modality at each visit (e.g. baseline rsfMRI to visit 3 rsfMRI), so that baseline seeds in DTI and rsfMRI space could be propagated to future visits. To transform seeds between volumes, transformation matrices output by align_epi_anat.py were applied using the AFNI routine 3dAllineate with nearest neighbor interpolation. Seeds used in the rsfMRI analysis included a subset of voxels used as seeds for DTI tracking, to ensure that only grey matter was included.

*Sensorimotor cortex (SMC).* The uni-manual tapping activation maps were used to identify the maximally activated voxel in the M1 motor region of the contralateral hemisphere (Fig 2A). To create the seeds used for DTI tracking, a 6 mm sphere was centered at this voxel in each hemisphere and was transformed from fMRI space to DTI space (Fig 2A, lower left). To create the seeds used for the rsfMRI analysis, a 9-voxel in-plane ROI was centered at the maximally active voxel in each hemisphere. ROIs were visually inspected and, if necessary,

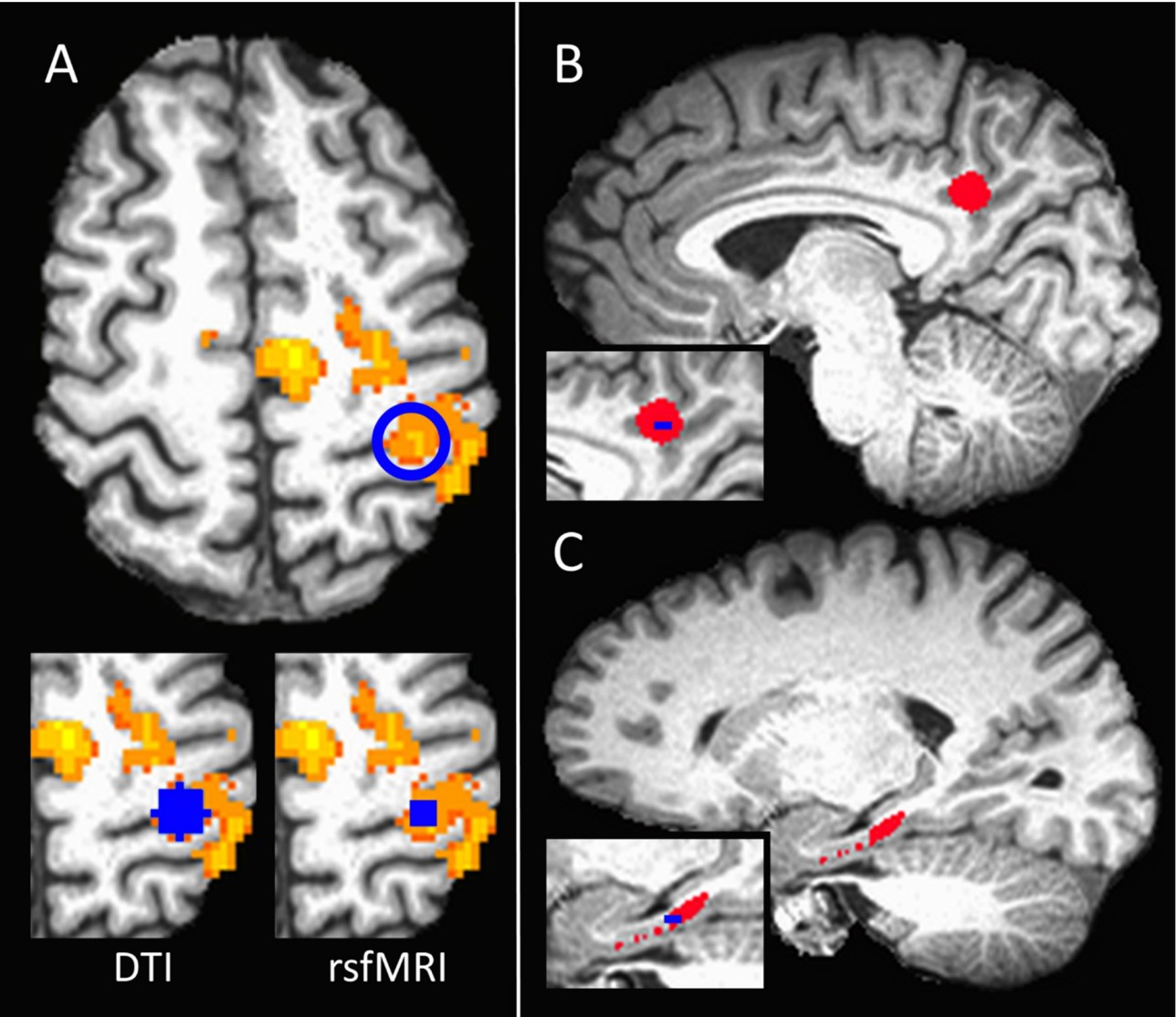

**Fig 2. Example of seed placement.** A) Seed placement for the left hemisphere sensorimotor cortex region of interest (ROI). The maximally activated voxel in the primary motor cortex during the right hand tapping functional MRI task is circled in blue. The lower left shows the 6mm sphere centered at that voxel used for the structural connectivity (DTI) analysis. The lower right shows the 9-voxel in plane seed used for the functional connectivity (rsfMRI) analysis. Anatomical ROIs for the B) posterior cingulate cortex and the C) entorhinal cortex. Insets show the cross section of the 9-voxel seed used for the rsfMRI analysis for each ROI.

manually eroded to ensure all voxels were located in gray matter and not in intragyral CSF (Fig 2A, lower right). Seeds were transformed from the task-activated volume to the rsfMRI volume.

*PCC-AMTL*. The PCC-AMTL seeds were defined using a combination of anatomical and functional data. First, the MPRAGE was linearly transformed to Talairach space using the AFNI routine @auto_tlrc. To define the PCC, a 6mm sphere was placed in each hemisphere at [−12 −42 36], according to coordinates reported in Greicius et al., 2003 [23] (Fig 2B). For the AMTL region, ROIs were drawn manually on the MPRAGE in Talairach space and included the entorhinal cortex and medial subiculum from coronal slices 10P-30P (Fig 2C). These ROIs

were transformed to original MPRAGE space, checked for accuracy, and transformed to the DTI and rsfMRI volumes. The transformed ROIs were used as the seeds for DTI tracking.

The rsfMRI time series was masked to include only grey matter voxels within the PCC and AMTL ROIs. Cross-correlation was used to identify the two voxels with the highest correlation between the PCC and AMTL [15]. For each ROI, this voxel was taken as the center of a 9-voxel in-plane seed, which functioned as the PCC and AMTL seeds for the functional connectivity analysis (Fig 2B and 2C, inset).

**Structural connectivity calculation.** The DTI metric, *sc*, was calculated using probabilistic tracking, performed using an analytic tracking algorithm based on partial differential equations and global and local constraints based on voxel level fiber orientation determined from the HARDI data [52]. The structural connectivity determination for each tracked pathway is described in detail in reference [13], and was calculated using in-house code. Briefly:

1. FOD determination: For each voxel, the 71-direction diffusion data were used to determine the local fiber orientation distribution (FOD) function [53]. This function determines the local probability for propagation of tracks from the seed (PCC and left SMC) to the target (AMTL and right SMC). Voxel-level probabilities for belonging to the pathway were then determined. Fig 3 shows thresholded tracks in a representative participant.

2. DTI determination: Tensor values of water diffusion were calculated for each voxel using a standard log-linear least squares method [54]. The tensor was diagonalized and the scalar measures of tissue microstructure (axial diffusivity (AD), radial diffusivity (RD), mean diffusivity (MD), and fractional anisotropy (FA)) were calculated for each voxel.

3. Pathway dependent diffusion measures were calculated using the track probability map, the scalar diffusion values, and a white matter mask [35]. Each measure <D> is calculated according to:

$$\langle D \rangle = \sum_v D(v) \cdot w(v) \cdot WM(v) \qquad\qquad Eq4$$

where the D(v) is the particular tensor-based value of interest (e.g. FA) at voxel v and w(v) is a so-called track probability map, in which the value of each voxel equals probability of track membership generated by the probabilistic tractography algorithm for that voxel. WM is a mask that is set to one for voxels determined to be mostly white matter from the Freesurfer segmentation. The result is $\langle FA \rangle$, $\langle MD \rangle$, $\langle AD \rangle$, and $\langle RD \rangle$ for every participant. Based on our prior work, we took pathway averaged RD to be our measure of structural connectivity, or *sc*.

**Functional connectivity calculation.** Using in-house code implemented in Matlab R2018b, the rsfMRI metric, *fc*, was calculated in the following manner:

1. A reference time series was calculated from the linearly detrended arithmetic average of the nine seed voxels. For the PCC-AMTL pathway, the bilateral PCC seeds were used. For the transcallosal motor pathway, the left SMC seed was used.

2. A whole-brain voxel-wise cross correlation map was calculated for each pathway of interest using the reference time series, resulting in three whole-brain rsfMRI correlation maps: left hemisphere SMC and right and left hemisphere PCC.

3. To account for individual differences in global signal, each correlation map was converted to a Student's t map. For each Student's t map, the whole-brain distribution was normalized to unit variance and zero mean [46]. The mean and variance from each distribution was used to convert the Student's t to a z-score.

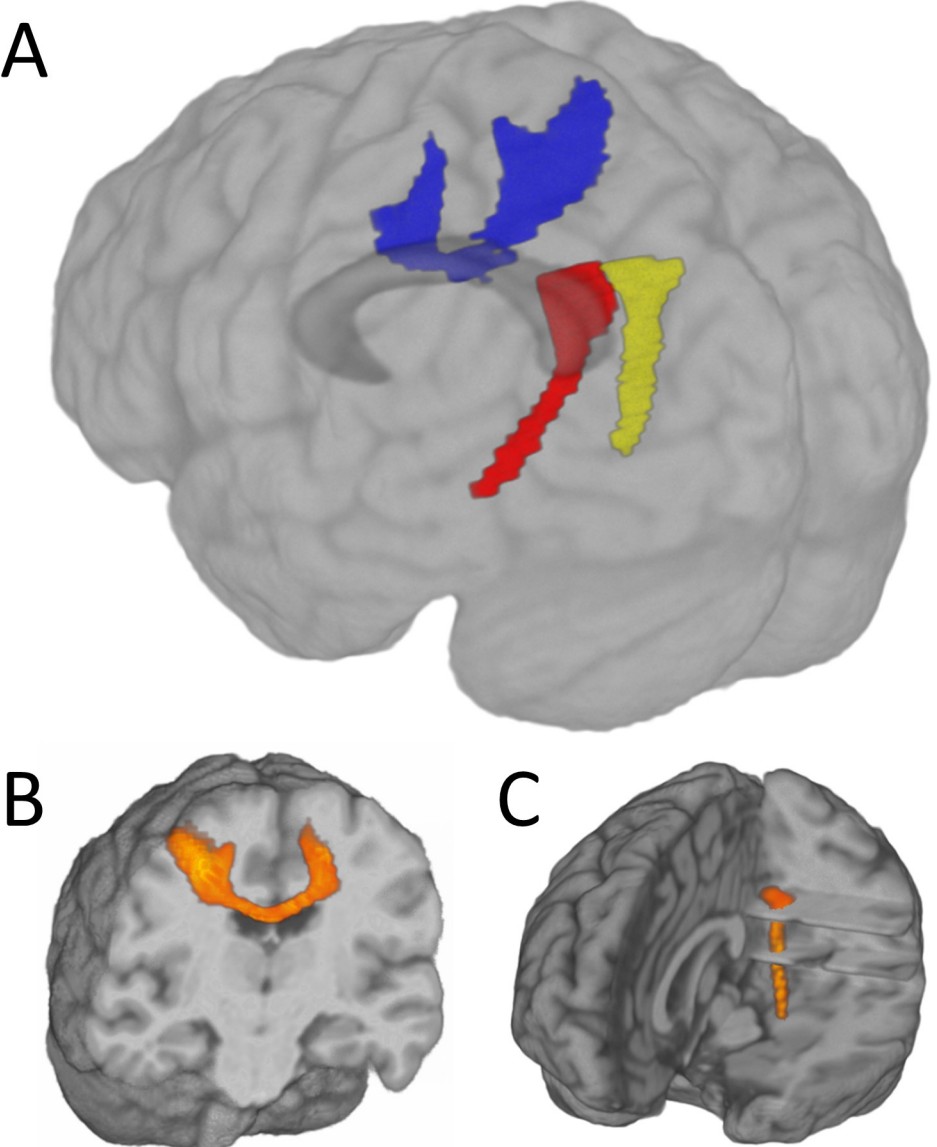

**Fig 3. Example of probabilistic tracking.** Individual subject probabilistic tracking results. A) Blue = transcallosal SMC, red = left hemisphere PCC-AMTL, yellow = right hemisphere PCC-AMTL track. The bottom row shows additional anatomical detail for B) the SMC and C) the left PCC-AMTL tracks. Displayed tracks are count-thresholded surfaces derived from probabilistic maps described in the text. SMC = primary sensorimotor cortices; PCC-AMTL = posterior cingulate cortex-antero-mesial temporal lobe.

4. The metric $fc$ was calculated for each pathway by taking the average z-score of the nine voxels in the target seed. For each hemisphere's PCC-AMTL pathway, the corresponding AMTL seed was used. For the transcallosal motor pathway, the right SMC ROI was used.

## Statistical analysis

To assess group differences in baseline and slope of imaging measures and SFCI and component measures, data from MS participants and controls at baseline and the final visit were entered into a linear mixed-effects (LME) analysis using R [55]. The model tested the impact

of group and the effect of time using visit, with education and scanner entered as covariates: [group * visit + scanner + education]. To assess group differences in the rate of change, a group × visit interaction was included. Random effects of subject were included for intercept and slope. A reduced model (omitting group: [visit + scanner + education]) was compared to the full model using the likelihood ratio test (LRT). The false discovery rate adjustment (FDR) was applied to correct for multiple comparisons [56].

Using all six visits, a second LME analysis assessed the impact of time in MS participants. The model assessed the linear effect of time using visit, with education, scanner, disease duration, and sex entered as covariates: [visit + scanner + education + disease duration + sex]. Random effects of subject were included for intercept and slope. A reduced model (omitting visit: [scanner + education + disease duration + sex]) was compared to the full model using the LRT. The FDR was applied.

In participants with MS, the relationship of baseline SFCI and associated imaging measures to baseline and final visit behavioral measures were assessed using Pearson correlations. Measures of motor function (the 9 hole peg test and 25 ft. walk) were log-transformed. To determine if changes in imaging were related to behavioral measures, slopes of SFCI components and imaging measures taken from the LME analysis of time were correlated with behavioral measures taken at the final visit. The FDR was applied for each imaging measure. Potential covariates, including age, education, disease duration, and EDSS, were correlated with imaging behavioral measures, including measures of change. A measure was included as a covariate only if it showed a relationship to either imaging or behavioral measures.

## Results

### Sample description

A total of 25 MS participants and 12 controls were enrolled. Three participants (2 with MS and 1 control) withdrew from the study prior to completion. Throughout the course of the study, two participants with MS were administratively removed due to a Beck Depression Inventory score > 20, two controls were removed due to very low baseline cognitive performance, and one participant with MS was removed due to extreme motion in all baseline scans. The final sample consisted of 20 participants with MS and 9 controls.

One participant with MS had four rather than six time points in the final sample. This participant withdrew after the fourth visit, but completed the final visit cognitive evaluation at that time. The third time point was removed for a different participant with MS, because technical difficulties prevented the acquisition of DTI data at that visit. The remaining sample was composed of 117 MS and 18 control visits.

Demographics, cognitive performance, and disease characteristics are reported in Table 2. All participants were strongly right handed (Edinburgh score>80) [57]. The groups did not show differences in age or sex, but the MS group had a significantly lower education level (p = 0.046). The MS group showed a decrease in MSFC from baseline to the final visit (p = 0.002). One participant with MS showed a 2-point increase in EDSS, and two showed a 1-point increase. At baseline, the MS group scored lower than controls on the CVLT (p = 0.013), BVMT-DR (p = 0.019) and SDMT (p = 0.039). The SDMT was the only cognitive measure that showed a significant decline from baseline to final visit in the MS group (p = 0.011).

### Scanner change

The scanner used for this study was upgraded from a Trio to a Prisma during data collection. Three controls were scanned using the Trio at baseline and the Prisma at follow up. The

**Table 2. Participant demographics, cognitive performance, and clinical characteristics.**

| | MS | | HC | | MS v HC | MS v MS |
|---|---|---|---|---|---|---|
| | **Baseline** | **Final Visit** | **Baseline** | **Final Visit** | **Baseline p** | **Time p** |
| **Demographics** | | | | | | |
| N (males) | 20 (6) | - | 9 (3) | - | 0.938 | - |
| Age | 50.95 ± 6.8 | - | 48.11 ± 7.5 | - | 0.324 | - |
| Education | 14.10 ± 2.2 | - | 16.00 ± 2.4 | - | 0.046 | - |
| **Cognitive performance** | | | | | | |
| CVLT | 46.00 ± 7.9 | 42.65 ± 11.6 | 54.44 ± 7.7 | 58.11 ± 7.5 | 0.013 | 0.088 |
| BVMT | 46.20 ± 13.6 | 42.45 ± 12.9 | 53.33 ± 5.9 | 53.89 ± 7.7 | 0.145 | 0.217 |
| BVMT-DR | 46.65 ± 14.3 | 44.65 ± 13.7 | 59.22 ± 6.8 | 57.44 ± 9.3 | 0.019 | 0.579 |
| SDMT | 50.10 ± 11.7 | 45.45 ± 12.3 | 59.78 ± 9.5 | 57.89 ± 9.3 | 0.039 | 0.011 |
| D-KEFS | 11.53 ± 2.7 | 11.11 ± 3.4 | 13.22 ± 3.1 | 13.33 ± 2.5 | 0.145 | 0.282 |
| JLO | 24.20 ± 3.8 | 24.40 ± 4.2 | 26.67 ± 3.3 | 26.89 ± 3.7 | 0.102 | 0.711 |
| COWAT | 38.40 ± 13.0 | 38.85 ± 14.1 | 40.78 ± 15.1 | 44.44 ± 10.3 | 0.669 | 0.816 |
| **Clinical characteristics** | | | | | | |
| EDSS | 3.5 (2–6.5) | 4.0 (2.5–6.5) | - | - | - | 0.069 |
| MSFC | -0.264 ± 0.58 | -0.446 ± 0.58 | 0.586 ± 0.38 | 0.383 ± 0.41 | $7.3\times10^{-4}$ | 0.002 |
| DD | 20.5 (4–33) | - | - | - | - | - |
| Lesion vol. (ml) | 16.89 ± 13.1 | 17.24 ± 10.9 | - | - | - | 0.994 |
| 25 ft. walk (sec.) | 8.29 ± 5.6 | 7.57 ± 2.8 | 4.03 ± 0.5 | 4.15 ± 0.5 | 0.043 | 0.175 |
| 9HPd (sec.) | 29.00 ± 24.1 | 29.68 ± 18.6 | 18.38 ± 2.4 | 19.95 ± 3.0 | 0.203 | 0.703 |
| 9HPnd (sec.) | 25.06 ± 5.9 | 26.74 ± 5.7 | 20.43 ± 3.2 | 20.42 ± 2.3 | 0.039 | 0.028 |
| Disease course | 12 RRMS | - | - | - | - | - |
| | 3 PPMS | | | | | |
| | 5 SPMS | | | | | |

Age, education, cognitive performance measures, MSFC, lesion volume, 25 ft. walk, and 9HP report mean ± standard deviation. DD reports mean (range). EDSS reports median (range). BVMT = Brief Visuospatial Memory Test; BVMT-DR = Brief Visuospatial Memory Test, delayed recall; COWAT = Controlled Oral Word Association Test; CVLT = California Verbal Learning Test; D-KEFS = Delis–Kaplan Executive Function System; DD = disease duration; EDSS = Expanded Disability Status Scale; HC = healthy control; JLO = Judgment of Line Orientation Test; MS = Multiple Sclerosis; MSFC = Multiple Sclerosis Functional Composite; PPMS = primary progressive; RRMS = relapse remitting; SDMT = Symbol Digit Modalities Test; SPMS = secondary progressive; 9HPd = 9 hole peg, dominant hand; 9HPnd = 9 hole peg, non-dominant hand.

remainder of the controls were scanned using the Prisma at both visits. Of participants with MS, one was scanned using the Trio for visits 1–4 and seven were scanned using the Trio for visits 1–3. For these eight participants, the remaining scans were completed using the Prisma. All other participants with MS were scanned using the Prisma at all visits. Separately for each group, t-tests were used to assess differences in imaging and demographic measures between those scanned at baseline on the Trio or Prisma. Participants with MS with baseline scans on the Prisma showed a significantly longer disease duration than those on the Trio (p = 0.0124), and *fc* of the left PCC-AMTL was significantly lower at visit 2 (p = 0.0088). No other measures showed differences in either MS or controls.

## MRI quality assessment

As described above, one participant with MS failed the baseline motion criteria and was administratively withdrawn from the study. On the basis of the criteria described for the remaining scans, rsfMRI scans from 20 participants with MS (20/117 scans) and 0 controls (0/18 scans) were inspected for evidence of motion-related artifact, including rings of correlation

around the outside of the head, correlation in the ventricles, and rapid correlation pattern changes from slice to slice (a consequence of motion in an interleaved style acquisition). Three connectivity scans (a single time point in each of three participants with MS) showed severe motion-related artifacts and were excluded from further analysis. Mean TDzmean and TDzmax were not significantly different between groups or across time in the remaining 114 connectivity scans.

## Group differences

Baseline group-averaged rsfMRI maps are shown in Fig 4 ($p < 1.0 \times 10^{-4}$, cluster size threshold = 60 voxels). The LME assessing the effect of group (MS vs. control) showed no differences in left PCC-AMTL and SMC *fc* at baseline or over time (Table 3; Fig 5A). Right PCC-AMTL showed effects of group*visit (p = 0.0104), but the model comparison did not survive FDR correction (p = 0.0241). The group analysis did not show a significant effect of education or scanner for any *fc* measure. The only *sc* measure to show a group difference was the SMC (p = 0.0054), showing an effect of group (p = 0.0115), but not of group*visit interaction, education, or scanner (Table 3; Fig 5B).

Table 3 shows the results of the LME analysis of MS vs. control differences for the SFCI and component measures. The full model (including "group") was significant for total SFCI (p = 0.0088) and $Z_{motor}$ (p = 0.0128), although the group and group*visit interaction terms did not reach significance (Fig 5C).

## Impact of time

In participants with MS, the LME assessing the impact of time showed a reduction in all *fc* measures (p < 0.05), but the change in SMC did not survive FDR correction (p = 0.0472; Table 4; Fig 6A). The time analysis did not show an effect of education, disease duration, or sex for any *fc* measure, although left PCC-AMTL showed an effect of scanner (p = 0.0172). In controls, paired t-tests of baseline and follow-up *fc* measures showed no significant differences (PCC-AMTL, left p = 0.0809; PCC-AMTL, right p = 0.5907; SMC p = 0.3870). None of the *sc* measures showed a significant impact of time in the MS participants (Table 4; Fig 6B). Paired t-tests showed no differences between baseline and follow-up *sc* measures in controls (PCC-AMTL, left p = 0.1874; PCC-AMTL, right p = 0.9430; SMC p = 0.9123). In participants with MS, BPF did not change over time. LV showed a weak increase over time, although this did not survive FDR correction (p = 0.0437; Table 4; Fig 6E).

The SFCI and the $Z_{motor}$ component declined over time in MS (p < 0.03), independently of scanner, education, disease duration, and sex (Table 4; Fig 6C and 6D).

## Imaging and behavioral measures

Tables 5 and 6 show Pearson correlation coefficients between imaging and behavioral measures in participants with MS. In the interest of space, only behavioral measures that showed a significant relationship to at least one imaging measure are included in the tables. One participant was an outlier (greater than 3 standard deviations from the mean) on the log-transformed 9 hole peg test score at both baseline and final visit. One participant was an outlier on the log-transformed 25 ft. walk score at baseline. Correlations with these measures excluded the outlier values. Age, education, disease duration, and EDSS were not significantly related to imaging or cognitive measures and were not included as covariates in the following analyses. The three participants that showed a 1-point or greater increase in EDSS did not show differences in imaging measures compared to the rest of the sample.

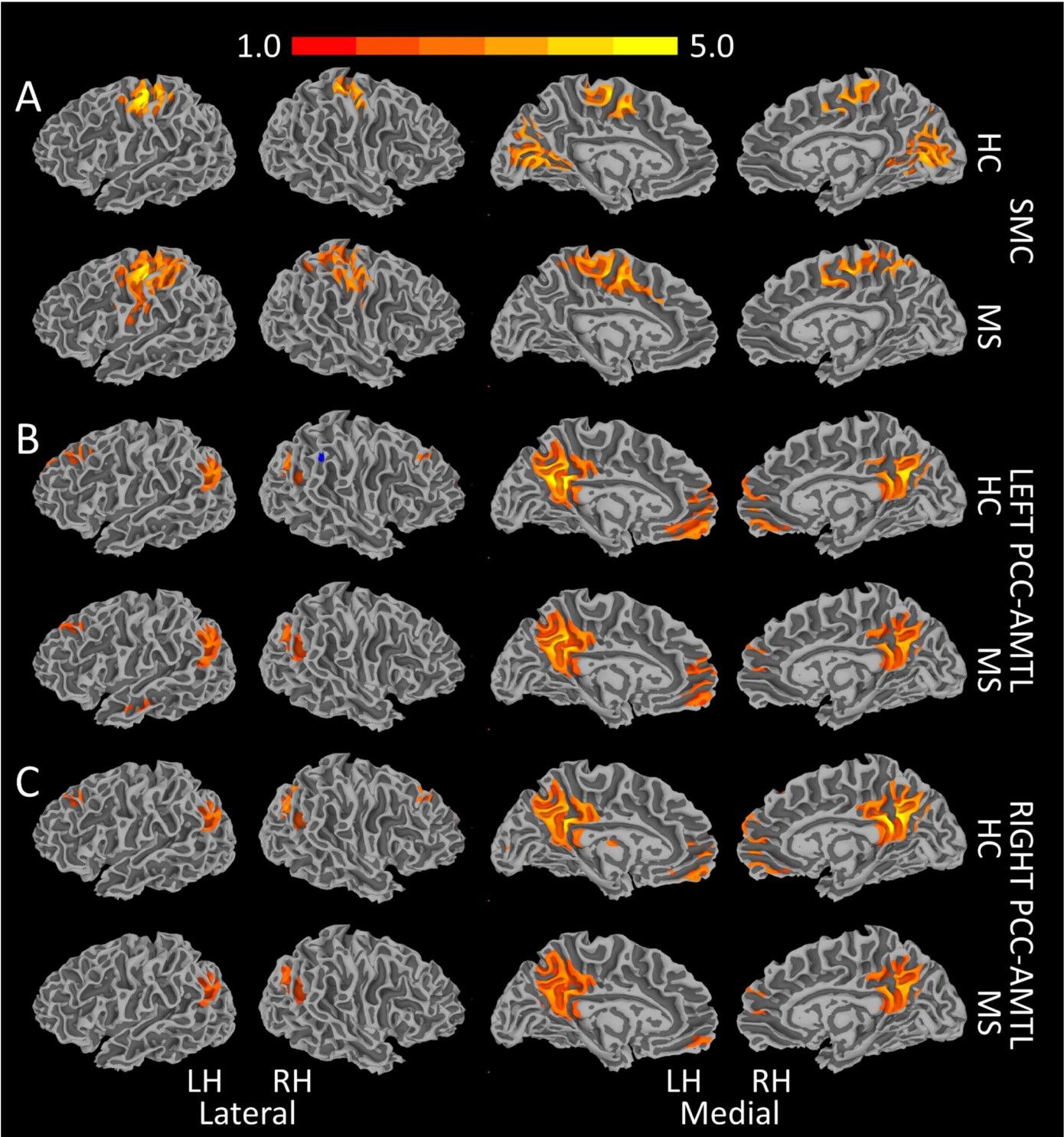

**Fig 4. Thresholded group-averaged functional connectivity maps.** Single voxel threshold is $1\times10^{-4}$ with a cluster requirement of 60 voxels. A. Transcallosal connectivity of the SMC. PCC-AMTL connectivity on the left (B) and right (C). HC = healthy control; LH = left hemisphere; MS = Multiple Sclerosis; PCC-AMTL = posterior cingulate cortex-antero-mesial temporal lobe; RH = right hemisphere; SMC = primary sensorimotor cortices.

**Table 3. Results of the LRT for group differences from baseline to the final visit in SFCI components and associated measures.**

| | Controls vs. MS | | | |
|---|---|---|---|---|
| | **LR** | **LRT $p$** | **group $p$** | **group*visit $p$** |
| **SFCI component** | | | | |
| SFCI | **9.48** | **0.0088** | 0.0707 | 0.0814 |
| $Z_{cog}$, left | 2.98 | 0.2255 | 0.4617 | 0.3075 |
| $Z_{cog}$, right | 7.22 | 0.0269 | 0.1992 | 0.1056 |
| $Z_{motor}$ | **8.72** | **0.0128** | 0.0603 | 0.1632 |
| **Imaging measures** | | | | |
| $fc$ PCC-AMTL, left | 1.40 | 0.4963 | 0.3473 | 0.3252 |
| $fc$ PCC-AMTL, right | 7.45 | 0.0241 | 0.1200 | 0.0104 |
| $fc$ SMC | 0.97 | 0.6155 | 0.4357 | 0.3846 |
| $sc$ PCC-AMTL, left | 3.46 | 0.1775 | 0.1489 | 0.5577 |
| $sc$ PCC-AMTL, right | 5.27 | 0.0718 | 0.0323 | 0.7381 |
| $sc$ SMC | **10.45** | **0.0054** | **0.0115** | 0.1820 |

The LRT was performed on the full model vs. the reduced model, which omits group. Values in bold survived FDR correction. $fc$ = functional connectivity; LR = likelihood ratio; LRT = likelihood ratio test; MS = Multiple Sclerosis; PCC-AMTL = posterior cingulate cortex-antero-mesial temporal lobe; $sc$ = structural connectivity; SFCI = Structural and Functional Connectivity Index; SMC = primary sensorimotor cortices; $Z_{cog}$ = SFCI cognitive component; $Z_{motor}$ = SFCI motor component.

Table 5 reports correlation coefficients between baseline imaging measures and behavioral performance at baseline and the final visit. In all tracks of interest, lower baseline $sc$ was associated with higher SDMT (p < 0.0007) and MSFC (p < 0.0007) scores at baseline and with higher SDMT (p < 0.006), MSFC (p < 0.03), and COWAT (p < 0.03) scores at the final visit. Baseline SDMT and non-dominant hand 9 hole peg score were the only measures associated with $Z_{motor}$ (p < 0.03). Other baseline SFCI measures were positively associated with baseline SDMT (p < 0.005) and MSFC (p < 0.01) scores, with higher SFCI related to higher scores. Similar positive relationships were observed between baseline SFCI measures and performance on the SDMT (p < 0.02) and COWAT (p < 0.01) at the final visit, although only right $Z_{cog}$ score was related to MSFC (p = 0.001) at the final visit.

Table 6 reports correlation coefficients between the change in imaging measures, represented by slopes calculated from the LME analysis of time, and behavioral performance at final visit. In the left PCC-AMTL and SMC, the change in $sc$ was negatively related to SDMT (p < 0.02), COWAT (p < 0.01), and MSFC (p < 0.02), such that an increase in $sc$ was associated with lower behavioral scores at the final visit. All SFCI slopes were positively related to SDMT (p < 0.02), MSFC (p < 0.03), and, with the exception of $Z_{motor}$, COWAT (p < 0.02), indicating that an increase in SFCI was related to higher behavioral scores at the final visit. Similarly, the change in BPF was positively related to MSFC (p < 0.005). Both total SFCI and $Z_{motor}$ were negatively related to the non-dominant hand 9 hole peg score (p < 0.03), so that a larger decrease in SFCI was associated with longer performance times at the final visit.

## Discussion

This work demonstrates that a combined structural and functional connectivity metric can be a sensitive tool for identification of clinical impairment in a neurological disease such as MS. The SFCI was constructed to probe neural pathways involved in symptomatic domains of MS, using complementary measures that represent different aspects of disease. For example, the SFCI and the $Z_{motor}$ component showed a difference between the MS and control groups and

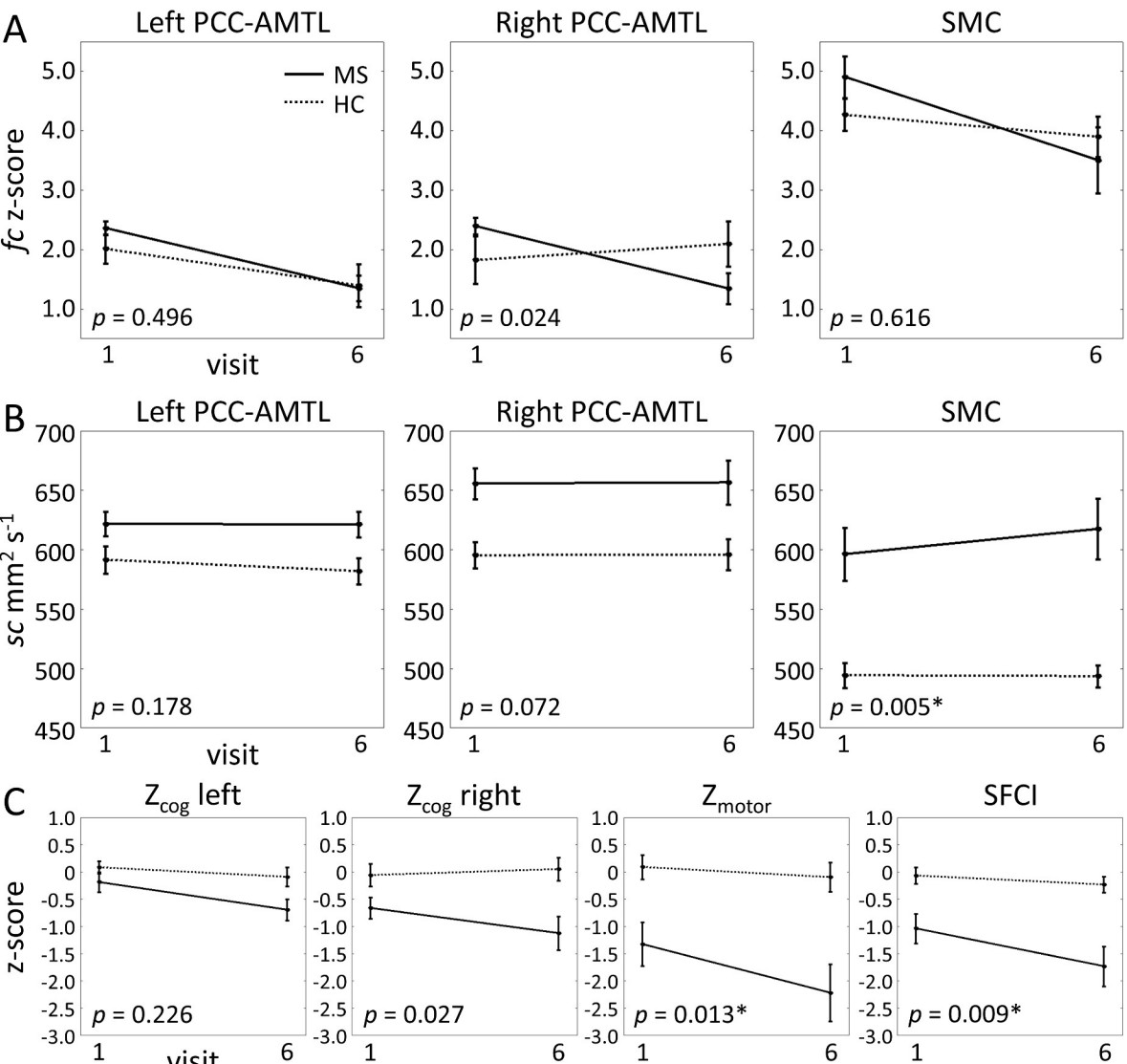

**Fig 5. Group-averaged imaging measures.** Group-averaged imaging measures plotted for MS and controls at baseline (1) and final visit (6). Error bars represent standard error. LRT *p*-values are included in each plot. A. rsfMRI (*fc*), B. DTI (*sc*), and C. SFCI component measures in the tracks of interest. * denotes significant result. HC = healthy controls; see Table 3 for additional abbreviations.

a change over time in participants with MS. Our *fc* measures showed no group differences at baseline, but demonstrated a significant decline over two years in MS participants. Conversely, non-significant baseline differences in *sc* measures drove group differences in SFCI.

Longitudinal effects on functional connectivity have not been studied widely in neurologic diseases [58], and the longitudinal relationship between functional and structural connectivity has not been well-characterized. Multiple investigators have reported a cross sectional relationship between structural and functional connectivity [59–65]. Some studies report a positive relationship [59–65], while others report a more complicated or inverse relationship [66, 67]. A recent longitudinal study of aging by Fjell et al. studied multiple pathways that were identified by either structural connectivity or functional connectivity [68]. They concluded that age-related changes in functional and structural connectivity are not necessarily strongly correlated, but that the highest positive change relationship is found in regions with a measured

**Table 4. The impact of time (visit) on SFCI components and associated measures in participants with MS.**

|  | MS | | |
| --- | --- | --- | --- |
|  | **LR** | **LRT $p$** | **visit $p$** |
| **SFCI component** | | | |
| SFCI | **6.67** | **0.0098** | **0.0121** |
| $Z_{cog}$, left | 2.07 | 0.1505 | 0.1609 |
| $Z_{cog}$, right | 0.70 | 0.4033 | 0.4192 |
| $Z_{motor}$ | **5.37** | **0.0205** | **0.0229** |
| **Imaging measures** | | | |
| $fc$ PCC-AMTL, left | **4.95** | **0.0261** | **0.0288** |
| $fc$ PCC-AMTL, right | **6.61** | **0.0102** | **0.0084** |
| $fc$ SMC | 3.94 | 0.0472 | 0.0408 |
| $sc$ PCC-AMTL, left | 0.46 | 0.4976 | 0.5344 |
| $sc$ PCC-AMTL, right | 1.07 | 0.3020 | 0.3030 |
| $sc$ SMC | 3.66 | 0.0556 | 0.0576 |
| Lesion volume | 4.07 | 0.0437 | 0.0504 |
| BPF | 1.44 | 0.2290 | 0.2121 |

The LRT was performed on the full model vs. the reduced model, which omits visit. Values in bold survived FDR correction. BPF = brain parenchymal fraction; see Table 3 for additional abbreviations.

functional connection. We interpret this study to imply that the existence of a structural path-way does not necessarily imply a functional connection, but a functional connection does imply a structural connection.

In MS participants, relationships between SFCI and behavioral measures were driven by *sc*, with all pathway measures related to the MSFC and its component measure, the SDMT. Although baseline SFCI was related to both baseline and longitudinal measures, the change in SFCI was also significant, so that those participants scoring lower on behavioral measures at follow up were more likely to show a decline in SFCI over time. The fact that no significant relationships were found between memory measures and imaging measures of the PCC-AMTL was unexpected, as previous work [19] found that RD was significantly related to both SDMT and BVMT in MS. A possible explanation is that the prior study used a DTI acqui-sition that had higher spatial resolution and was tailored to target the PCC-AMTL pathway. Although the SDMT has been associated with track-specific changes in MS [69, 70], it has also been associated with whole brain measures such as atrophy and lesion volume [71]. Given the association of the PCC-AMTL to memory function, future work is required to clarify the lack of relationship seen here.

Clinical trials of MS treatments typically do not include structural and functional connec-tivity measures as outcomes, more commonly reporting LV and brain atrophy. Although LV has been previously associated with episodic memory in MS [72], here LV showed no change over the course of the study and showed no relationship to cognitive measures. Similarly, base-line BPF was not related to cognitive measures. The magnitude of BPF change was related to MSFC at the final visit, although changes in SFCI, $Z_{motor}$, and *sc* SMC showed similar or stron-ger relationships.

In this study, we show that a combined metric of structural and anatomic connectivity in symptomatic domains of MS is sensitive to the longitudinal progression of disease over two years. The SFCI was constructed as a single, MRI-based metric that can be easily implemented as an outcome measure in clinical trials. The metric is z-scored and changes have, in principle,

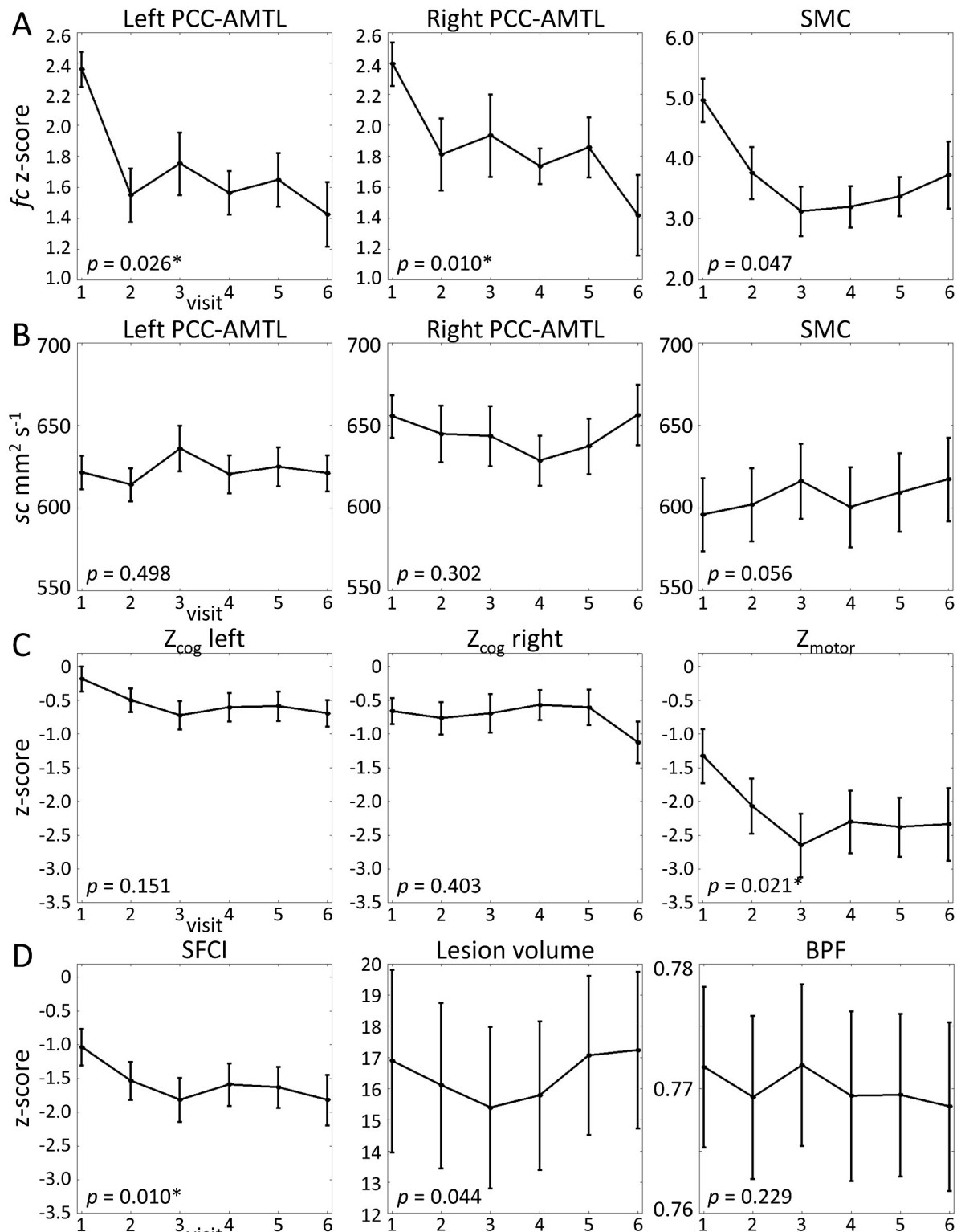

**Fig 6. Averaged imaging measures in participants with MS as a function of visit.** Error bars represent standard error. LRT *p*-values are included in each plot. A. rsfMRI (*fc*), B. DTI (*sc*), C. SFCI components, and D. SFCI in the tracks of interest. E. Whole-brain imaging measures. *denotes significant result. BPF = brain parenchymal fraction; see Table 3 for additional abbreviations.

**Table 5. Pearson correlation coefficients between baseline imaging measures and behavioral measures at baseline and the final visit in 20 participants with MS.**

| | Baseline visit | | | | Final visit | | |
|---|---|---|---|---|---|---|---|
| | SDMT | COWAT | MSFC | 9HPnd | SDMT | COWAT | MSFC |
| **SFCI component** | | | | | | | |
| SFCI | **0.645**\*\* | 0.453 | **0.590**\* | **-0.519**\* | **0.545**\* | **0.564**\* | 0.504 |
| $Z_{cog}$, left | **0.696^** | **0.568**\* | **0.670**\*\* | -0.379 | **0.559**\* | **0.675**\*\* | 0.465 |
| $Z_{cog}$, right | **0.702^** | **0.510**\* | **0.566**\* | -0.257 | **0.774^** | **0.719^** | **0.709^** |
| $Z_{motor}$ | **0.513**\* | 0.359 | 0.489 | **-0.549**\* | 0.366 | 0.407 | 0.340 |
| **Imaging measures** | | | | | | | |
| *fc* PCC-AMTL, left | 0.390 | 0.454 | 0.475 | -0.040 | 0.177 | 0.383 | 0.095 |
| *fc* PCC-AMTL, right | 0.112 | 0.291 | 0.025 | 0.379 | 0.334 | 0.288 | 0.231 |
| *fc* SMC | -0.371 | -0.038 | -0.356 | -0.193 | -0.437 | -0.164 | -0.478 |
| *sc* PCC-AMTL, left | **-0.696^** | **-0.516**\* | **-0.632**\*\* | 0.446 | **-0.612**\*\* | **-0.674**\*\* | **-0.514**\* |
| *sc* PCC-AMTL, right | **-0.702^** | -0.441 | **-0.588**\* | 0.397 | **-0.704^** | **-0.661**\*\* | **-0.667**\*\* |
| *sc* SMC | **-0.720^** | -0.393 | **-0.687^** | 0.477 | **-0.599**\* | **-0.507**\* | **-0.613**\*\* |
| Lesion volume | -0.017 | 0.085 | -0.232 | 0.330 | -0.552 | 0.179 | 0.284 |
| BPF | 0.090 | -0.157 | 0.212 | -0.161 | 0.550 | -0.176 | -0.226 |

Uncorrected p-values

\* < 0.03

\*\* < 0.005

^ < 0.001. Values in bold survived FDR correction. BPF = brain parenchymal fraction; COWAT = Controlled Oral Word Association Test; MSFC = Multiple Sclerosis Functional Composite; SDMT = Symbol Digit Modalities Test; 9HPnd = 9 hole peg, non-dominant hand; see Table 3 for additional abbreviations.

**Table 6. Pearson correlation coefficients for the slope of imaging measures related to behavioral measures at the final visit in 20 participants with MS.**

| | Final Visit | | | |
|---|---|---|---|---|
| | SDMT | COWAT | MSFC | 9HPnd |
| **SFCI component** | | | | |
| SFCI | **0.603**\*\* | **0.568**\*\* | **0.665^** | **-0.518**\* |
| $Z_{cog}$, left | **0.534**\* | **0.574**\*\* | **0.522**\* | -0.284 |
| $Z_{cog}$, right | **0.635^** | **0.558**\* | **0.617**\*\* | -0.165 |
| $Z_{motor}$ | **0.565**\*\* | 0.461 | **0.738^** | **-0.529**\* |
| **Imaging measures** | | | | |
| *fc* PCC-AMTL, left | 0.180 | 0.143 | 0.127 | 0.172 |
| *fc* PCC-AMTL, right | 0.245 | 0.285 | 0.145 | 0.439 |
| *fc* SMC | -0.087 | 0.024 | -0.006 | -0.397 |
| *sc* PCC-AMTL, left | **-0.544**\* | **-0.598**\*\* | **-0.550**\* | 0.384 |
| *sc* PCC-AMTL, right | -0.328 | -0.252 | -0.261 | -0.036 |
| *sc* SMC | **-0.559**\* | **-0.560**\* | **-0.688^** | 0.378 |
| Lesion volume | -0.332 | -0.246 | -0.478 | **0.543**\* |
| BPF | 0.460 | 0.278 | **0.633**\*\* | -0.482 |

Uncorrected p-values

\* < 0.03

\*\* < 0.01

^ < 0.005. Values in bold survived FDR correction. BPF = brain parenchymal fraction; COWAT = Controlled Oral Word Association Test; MSFC = Multiple Sclerosis Functional Composite; SDMT = Symbol Digit Modalities Test; 9HPnd = 9 hole peg, non-dominant hand; see Table 3 for additional abbreviations.

a straightforward statistical interpretation. A decrease in the metric over time reflects declining pathway connectivity, which may be a surrogate for worsening disease. It is important to note that it was necessary to use a statistical model in this study to demonstrate statistically significant changes in the metric due to the major equipment change that occurred during the study. In a situation with stable equipment, changes in the metric have a simple statistical interpretation.

The proposed SFCI measure performed well in this sample, and the focus on specific pathways makes the interpretation straightforward. The pathway-based imaging measures used in this study have an advantage over other, region-based imaging measures in that it is not necessary to perform any spatial normalization. Although larger, anatomical-based landmarks were used as starting points for seed determination, the final seed regions were identified within-subject by brain function-related techniques, and all metrics were determined in the subject's native space. This eliminates errors and processing biases that can be introduced by spatial normalization across groups. On the other hand, this methodology is difficult to implement on a large scale, and would benefit from simplification.

The performance of the SFCI in this initial test is encouraging, but given our relatively small sample size, it will require replication in a larger sample and will likely benefit from refinement. As only a few participants showed declines in function as measured by EDSS, we were unable to describe the relationship of SFCI to clinically relevant functional decline. Larger samples will also be required to break down SFCI performance by disease course or treatment. Finally, the methods used here are highly specialized and require considerable data processing effort. Wider applicability of the SFCI will require additional simplification and automation of data processing.

The pathways used here were chosen based on past work showing relationships between connectivity strength and cognitive and functional measures in MS. Although disease mechanisms differ, it is worth noting that the development of a pathway-based, domain-specific composite measure is not restricted to MS. This approach may be applicable in other neurologic diseases that have symptomatic domains that can be related to pathway or network connectivity impairment.

## Conclusion

We present evidence that combined structural and functional connectivity measures show significant decrease over the duration of a longitudinal study of Multiple Sclerosis. The combination of structural and functional connectivity along pathways implicated in specific domains of disability in MS resulted in a metric that was related to both disease status and functional decline. The SFCI demonstrated a significant difference between participants with MS and sex- and age- matched HC, as well as a significant two-year decline in participants with MS. Both baseline SFCI and the change over time were related to a measure of speed of processing after two years, and the change in SFCI was related to future MSFC. This indicates that combined structural and functional connectivity measures, tailored to domains of disability, can be sensitive biomarkers for the study of neurologic disease.

## Supporting information

**S1 Fig. ROC curves for measure 1 ($M_1$) for progression scenario 1.**
(TIF)

**S2 Fig. Progression scenario 1.** Area under the ROC curve for measures 1–3 ($M_1$, $M_2$, and $M_3$), nine time points.
(TIF)

**S3 Fig. Progression scenario 2.** Area under the ROC curve for measures 1–3 ($M_1$, $M_2$, and $M_3$), nine time points.
(TIF)

**S1 Text. Development of SFCI.**
(DOCX)

## Author Contributions

**Conceptualization:** Mark J. Lowe.

**Data curation:** Katherine A. Koenig, Erik B. Beall, Stephen M. Rao, Kunio Nakamura.

**Formal analysis:** Katherine A. Koenig, Erik B. Beall, Ken E. Sakaie, Kunio Nakamura, Mark J. Lowe.

**Funding acquisition:** Mark J. Lowe.

**Investigation:** Katherine A. Koenig, Erik B. Beall, Daniel Ontaneda, Lael Stone, Stephen M. Rao, Stephen E. Jones, Mark J. Lowe.

**Methodology:** Katherine A. Koenig, Erik B. Beall, Ken E. Sakaie, Stephen M. Rao, Kunio Nakamura, Mark J. Lowe.

**Project administration:** Mark J. Lowe.

**Resources:** Daniel Ontaneda, Lael Stone.

**Software:** Erik B. Beall, Ken E. Sakaie, Kunio Nakamura.

**Supervision:** Mark J. Lowe.

**Visualization:** Katherine A. Koenig, Ken E. Sakaie.

**Writing – original draft:** Katherine A. Koenig, Erik B. Beall, Ken E. Sakaie, Mark J. Lowe.

**Writing – review & editing:** Katherine A. Koenig, Ken E. Sakaie, Daniel Ontaneda, Mark J. Lowe.

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
