## [Decision Letter · Decision Letter 0]

30 Nov 2020

PONE-D-20-32068

Evaluation of a connectivity-based imaging metric that reflects functional decline in Multiple Sclerosis

PLOS ONE

Dear Dr. Koenig,

Thank you for submitting your manuscript to PLOS ONE. After careful consideration, we feel that it has merit but does not fully meet PLOS ONE’s publication criteria as it currently stands. Therefore, we invite you to submit a revised version of the manuscript that addresses the points raised during the review process.

In addition to the comments from the Reviewers, please also address the following:

- Carefully check all references, as at least one is improperly formatted/referenced (ref. 34) 

We look forward to receiving your revised manuscript.

Kind regards,

Niels Bergsland

Academic Editor

PLOS ONE

Journal Requirements:

Reviewers' comments:

Reviewer's Responses to Questions

**Comments to the Author**

1. Is the manuscript technically sound, and do the data support the conclusions?

Reviewer #1: Partly

Reviewer #2: Yes

2. Has the statistical analysis been performed appropriately and rigorously? 

Reviewer #1: Yes

Reviewer #2: Yes

3. Have the authors made all data underlying the findings in their manuscript fully available?

Reviewer #1: No

Reviewer #2: No

4. Is the manuscript presented in an intelligible fashion and written in standard English?

Reviewer #1: Yes

Reviewer #2: Yes

5. Review Comments to the Author

Reviewer #1: This very interesting paper looks at pathway-based MRI measures of structural and functional connectivity to construct the motor and cognition dysfunction by a composite imaging metric in MS. The topic is of high interest, given the lack of knowledge on relations between structure and function in relation to clinical deficits. Results are related to disability and cognitive dysfunction. Technically the setup is alright, but the descriptive indicators are too strict, headings and theirs order are confusing. Thresholds for the DTI analysis are missing which effects the correlates between the functioning. Hypothesis is misleading. MS has more complex relationships between pathways and functioning than AD. So, the selected pathway may not represent the memory domain functioning only in MS. Likewise, the current study also showed SDMT correlation as well, which is one the most common cognitive test in MS to indicate the processing speed functioning. The recommendations regarding the correction of these descriptions are attached below.

Abstract:

- Well-written

- Add MS cohort

- Perhaps indicate “2-year longitudinal study at four month intervals” info.

Introduction

- Line 46-48 “We reported reduced resting state functional magnetic resonance imaging (rsfMRI) between the bilateral primary motor cortices in patients with MS as compared to controls” is wrong expression. Perhaps rephrase to “reduced connectivity assessed by rsfMRI”?

- Line 48-50 “A follow-up study focused on both structural and functional connectivity, showing that diffusion tensor imaging (DTI) measures in the transcallosal motor pathway are correlated with rsfMRI of the primary sensorimotor cortices (SMC) in MS” How was this correlation? Perhaps indicate “The change of DTI and rs-fMRI measures were positively correlated over …”

- Line 69-72 “The cognitive component of the MSFC is based on the Paced Serial Addition Test (PASAT) [14], or, more recently, the Symbol Digit Modalities Test (SDMT). The metric proposed here was constructed as an analog to these composite measures of neurologic deficit.” Is this indicating the current study? This is very confusing. In the method section, different kind of tests are existed, but no PASAT.

- Line 73-75 “We focus on two common domains of disability in MS, motor function and memory,..” Why memory only? Several cognitive tests are given in the method and results sections. Conclusion also does not indicate this info. Perhaps the term was going to be “cognition” or “cognitive function”? In addition, even if the main idea was to track the memory, PASAT, BVMT and etc would have been chosen to specify, but not SDMT. SDMT more indicates processing speed.

On the other hand, PCC-AMLT is a well-known memory pathway in AD. However, MS has more complicated pathways that are presented by several cognitive dysfunction in it. Based on the results of the current study, it seems that this pathway is related with memory (COWAT) and processing speed (SDMT) domains. Therefore, instead of memory only, representing as “motor and cognitive functions” would be better.

- Line 73-75: The hypothesis is poorly introduced. Referring the previous work does not help to understand the aim. The citation for the ROI selection for the specific domain should be explained in previous paragraph. Than the exact functions and pathways that are going to be investigated in this study should be described in the hypothesis-paragraph. Such as “therefore, we investigated this this this pathway and its relationship with this this functions using SFCI…. over 2 years..”??

Theory

- Line 89-90. Why the metric will decrease with increased disability? It is a fact that both adaptive and maladaptive responses can occur in MS.

- Line 99. Is the “sample means” indicates the mean of the global connectivity for each patient? The equation should be explained better. Such as for Eq.2: fcpop indicates global or individual functional connectivity of healthy controls, �cpop represents the standard deviation of global or individual functional connectivity of healthy controls and Zmotor indicates the effect size of the motor SFC…

- Line 106. Please introduce the terms in the Eq. 3. What kind of notation is the L,R etc.

- Line 109. Again please explain the notation.

Materials and methods:

- Line 118. The term “Data acquisition” sounds like collecting imaging data etc... Perhaps reword as “Participants”?

In this section, it is written that the study includes 25 MS and 12 HC but in the abstract it is written 20 MS and 9 HC. Indication of the only final numbers of the participant (after withdraw) would be better to follow the article. Another way to describe better would be adding the inclusion/exclusion criteria’s and writing the final number of participants was determined based on these criteria’s following neuropsychological examination given in results.

Ethical committee number and chair should be written.

- Sample size is too strict. Is the Power analysis done? If it is not within the acceptable range the calculations should be repeated by applying something like Bootstrapping.

- Line 139. Table 1. SDMT is rather processing speed. Processing speed and working memory functioning should not be combined under the same section titled as “memory”. SDMT and D-KEFS do not represent the components of the memory domains and rather PASAT would have been involved in the neuropsychological tests to show the working memory functioning as described in the introduction.

- Line 148: Numerical indication of the slices would be easier to read: 176 axial slices.

- Line 173. Only “MRI post- processing” heading misleads the readers. This part includes both pre and post processing for the current version. Perhaps the heading would be “Processing of the MRI data”?

Other MRI Measures

- Line 200-203. Please specify the lesion filling procedures/tolls etc. more detailed.

- Line 249: What are the exact values for the thresholding of the structural connectivity construction? FOD cut off, number of fibers etc.?

Results

- Line 335: Table 2 should include the lesion volume information.

- Descriptive statistics that are represented in the Table3&4 are hard to follow in the text. Combining the functional and structural connectivity sections would be better for following the results in the tables that belong to both sections.

- Line 417-418: “In the interest of space, only behavioral measures that showed significant relationships are included.” Which relationship? Did you mean that only those that show a significant correlation between at least one of the SFCI components and one of the neuropsychological tests are given in the table?

- Line 431-446: Instead of the terms “positively/negatively”, explaining like “lower performance by SDMT or etc related with the lower/stronger/higher connectivity” may be easier to read.

Discussion

-Line 448-449: “This work demonstrates that a combined structural and functional connectivity metric can be a sensitive tool for identification of neurological disease” there is no other type of neurological disease that shows different patterns etc than MS or identify MS. So this sentence may be confusing. Rather it can be “This work demonstrates that a combined structural and functional connectivity metric can be a sensitive tool for identification of clinical impairment in neurological disease such as MS.”

-Methodological limitations must be discussed

-Line 497-500: I respectfully disagree with the authors that normalization is not necessary to compensate bias field effect (i.e noise) either it is region based or not. Before the calculation of the ROIs, registration to a standard atlas is mandatory which the current study also involved registration. This issue should be discussed more. Perhaps, exemplify the order of the analysis for this study compared to other studies.

Reviewer #2: The authors propose a composite MRI index, made out from MRI connectivity metrics relative to motor and cognitive domains, to relate to functional status and its decline in MS, tested in a 2-year longitudinal dataset.They focus on the transcallosal motor pathway connectivity and the connectivity mediated by the posterior cingulum bundle.

The authors theorize three different models to obtain the index and they test them both on simulated and real data.

line 150 : 256 x 128 might be a matrix error? ( siemens has this 50% distance factor - or gap- of the voxel on MPRAGE sequences in the direction of acquisition , but this does not change the matrix). I suppose also axial acquisition has to be verified, because 176 slices is typical of sagittal acquisitions.

liines 148-171 : please keep same order in the parameter description of each sequence (e.g. slice number,, thickness, FOV, matrix, voxel size etc). Also please report the acquisition duration, confirm that this is the order of acquisition and describe experimental setting , paradigm , pre-scan instructions and training.

line 172 MRI data analysis:: please provide all tools that you use in every step i.e. to apply Gaussian filter, fitting the boxcar model, the LV and on. I suppose that this can be provided as supplementary info. Also more info is needed for the experimental set, e.g. bite bar, physiological parameter acquisition. This is to ameliorate reproducibility.

line 219-220: How did you manually modify ROIs, e.g. by erosion of the voxels found in CSF?

line 230-242 : please add details for registration and transformation of images and refer to different spaces in a more explicit way: individual space can be functional , DTI, T13D or FLAIR. What kind of interpolation did you use to apply a transformation matrix to a flat ROI?

Fig3 : the 3D glass brain image appears blurred and does not permit a good evaluation of the localization.

line 265: by "cluster size 60" do you mean cluster size threshold of 60 voxels?please rephrase.

line 475-476: some grammar error, please rephrase

line 479 : rather than finding it is a missed relationship or no finding.

Conclusion:

This study can be included among the efforts in the advanced neuroimaging in neurological disease to identify connectivity biomarkers of disease status and progression. The small sample size may determine type I statistical error, as might be the case of the discussed missing PCC-AMTL association to memory. It is comprehensible that producing data for a power analysis at this point is too much to ask, therefore the value of this study can only be considered as in an exploratory study. It is valuable however for the longitudinal design, the right choice to focus on specific function-structure target, the sane methodology and for the correct admission and description of the limits. It is a valid scientific contribution in the search of viable imaging targets to obtain biomarkers that can translate into clinic, after validation in an adequately large study, so it should be considered as such.

It would be desired the availability, when published, of anonymized metadata of the group analysis, for example z-score maps and demographic, clinical and the other metrics that have been used, upon which to make a power analysis by anyone interested: itwould be an important addition for the usefullness of the paper and would fulfill the journal's policy and mission.

6. PLOS authors have the option to publish the peer review history of their article (what does this mean?). If published, this will include your full peer review and any attached files.

Reviewer #1: No

Reviewer #2: **Yes: **Nikolaos Petsas

---

## [Author Response · Author response to Decision Letter 0]

25 Feb 2021

Thank you to the reviewers for your thorough and insightful comments on our manuscript. We think the manuscript is stronger and more readable with your suggestions. Below, each reviewer comment is numbered, with our response following (indented).

Reviewer #1: 

Abstract

1. Add MS cohort

 The abstract was modified so that the description of the cohort is earlier in the paragraph. 

2. Perhaps indicate “2-year longitudinal study at four month intervals” info.

 This information has been added to the abstract. Please note that we corrected an error in the description of the study visits – the first year included four visits at 4 month intervals, and the second year included two visits spaced at 6 month intervals. This has been corrected in the text.

Introduction

3. Line 46-48 “We reported reduced resting state functional magnetic resonance imaging (rsfMRI) between the bilateral primary motor cortices in patients with MS as compared to controls” is wrong expression. Perhaps rephrase to “reduced connectivity assessed by rsfMRI”?

 This sentence was rephrased as suggested. 

4. Line 48-50 “A follow-up study focused on both structural and functional connectivity, showing that diffusion tensor imaging (DTI) measures in the transcallosal motor pathway are correlated with rsfMRI of the primary sensorimotor cortices (SMC) in MS” How was this correlation? Perhaps indicate “The change of DTI and rs-fMRI measures were positively correlated over …”

 We clarified that this was an inverse relationship.

5. Line 69-72 “The cognitive component of the MSFC is based on the Paced Serial Addition Test (PASAT) [14], or, more recently, the Symbol Digit Modalities Test (SDMT). The metric proposed here was constructed as an analog to these composite measures of neurologic deficit.” Is this indicating the current study? This is very confusing. In the method section, different kind of tests are existed, but no PASAT.

 This sentence was modified to clarify that the SFCI is designed to be used in a similar manner to the MSFC – as a composite measure of deficit. 

6. Line 73-75 “We focus on two common domains of disability in MS, motor function and memory,..” Why memory only? Several cognitive tests are given in the method and results sections. Conclusion also does not indicate this info. Perhaps the term was going to be “cognition” or “cognitive function”? In addition, even if the main idea was to track the memory, PASAT, BVMT and etc would have been chosen to specify, but not SDMT. SDMT more indicates processing speed.

On the other hand, PCC-AMLT is a well-known memory pathway in AD. However, MS has more complicated pathways that are presented by several cognitive dysfunction in it. Based on the results of the current study, it seems that this pathway is related with memory (COWAT) and processing speed (SDMT) domains. Therefore, instead of memory only, representing as “motor and cognitive functions” would be better.

 We used the term “memory” based on the primary function of the PCC-AMTL pathway. However, the reviewer makes a good point – in our previous work we found that PCC-AMTL connectivity was related to both BVMT and SDMT in MS, so is not exclusively memory-related. We use the more general term “cognitive dysfunction” repeatedly throughout the manuscript, and as a metric we hope the SFCI would capture multiple domains of cognitive dysfunction. We have modified the manuscript to use more general language, including renaming the memory component of the SFCI to the cognitive component of the SFCI. 

7. Line 73-75: The hypothesis is poorly introduced. Referring the previous work does not help to understand the aim. The citation for the ROI selection for the specific domain should be explained in previous paragraph. Than the exact functions and pathways that are going to be investigated in this study should be described in the hypothesis-paragraph. Such as “therefore, we investigated this this this pathway and its relationship with this this functions using SFCI…. over 2 years..”??

 We reconfigured the Introduction to, hopefully, flow more naturally to our hypothesis. We moved the EDSS/MSFC discussion to the second paragraph, to give examples of other composite metrics as we explain the development of the SFCI, an imaging-based composite metric. We also added verbiage to motivate the transcallosal motor pathway and explicitly stated the expected relationship between pathway and behavioral measures. 

Theory

8. Line 89-90. Why the metric will decrease with increased disability? It is a fact that both adaptive and maladaptive responses can occur in MS.

 The reviewer is correct that compensatory mechanisms can result in increased connectivity between some regions involved in a task with measured deficits, and there is substantial literature on these results. However, the pathways probed here are monosynaptic and the relationship between connectivity to these regions and clinical disability has been reported as inversely related (i.e. increased disability corresponds to decreased connectivity in these pathways). Based on prior findings, we expect this measure, as constructed, to decrease with disability. That said, as constructed, the metric will decrease with decreased connectivity, which is what we meant to say. We have modified the text accordingly and are grateful to the reviewer for pointing this out.

9. Line 99. Is the “sample means” indicates the mean of the global connectivity for each patient? The equation should be explained better. Such as for Eq.2: fcpop indicates global or individual functional connectivity of healthy controls, �cpop represents the standard deviation of global or individual functional connectivity of healthy controls and Zmotor indicates the effect size of the motor SFC… 

 Notation for the equation has been added.

10. Line 106. Please introduce the terms in the Eq. 3. What kind of notation is the L,R etc.

 Notation for the equation has been added. 

11. Line 109. Again please explain the notation.

 Notation for the equation has been added.

Materials and methods

12. Line 118. The term “Data acquisition” sounds like collecting imaging data etc... Perhaps reword as “Participants”?

 We added a section labeled “Participants” under the “Data acquisition” heading. We intended the “Data acquisition” heading to include the subsections of “Clinical and cognitive evaluation” and “MRI acquisition,” but agree with the reviewer that without a section labeled “Participants,” the intention of the “Data Acquisition” heading is not clear. 

13. In this section, it is written that the study includes 25 MS and 12 HC but in the abstract it is written 20 MS and 9 HC. Indication of the only final numbers of the participant (after withdraw) would be better to follow the article. Another way to describe better would be adding the inclusion/exclusion criteria’s and writing the final number of participants was determined based on these criteria’s following neuropsychological examination given in results.

 To clarify the sample size, we added the following sentence directly after stating the total enrollment: “Of this sample, twenty participants with MS and 9 healthy controls were included in the final data analysis (see Results: Sample Description for details).”

14. Ethical committee number and chair should be written.

 The protocol IRB number is now included. The chair of the Cleveland Clinic IRB is a rotating position that may be different upon publication. Therefore, rather than including a specific name, we elected to include the Federalwide Assurance number for the Cleveland Clinic IRB. This is now included.

15. Sample size is too strict. Is the Power analysis done? If it is not within the acceptable range the calculations should be repeated by applying something like Bootstrapping.

 Yes, a power analysis was completed during the project development phase of this study. This project was funded by the NMSS, and the sample sizes proposed in that grant application were based on a power analysis derived from preliminary data from prior studies that are discussed in the manuscript. The results we present are based on the data collected for that study.

16. Line 139. Table 1. SDMT is rather processing speed. Processing speed and working memory functioning should not be combined under the same section titled as “memory”. SDMT and D-KEFS do not represent the components of the memory domains and rather PASAT would have been involved in the neuropsychological tests to show the working memory functioning as described in the introduction.

 We modified the manuscript to focus on cognitive function rather than memory, as suggested by the reviewer in point 6. 

17. Line 148: Numerical indication of the slices would be easier to read: 176 axial slices.

 This has been modified.

18. Line 173. Only “MRI post- processing” heading misleads the readers. This part includes both pre and post processing for the current version. Perhaps the heading would be “Processing of the MRI data”?

 The heading has been changed to “MRI data processing.”

Other MRI Measures

19. Line 200-203. Please specify the lesion filling procedures/tolls etc. more detailed.

 No lesion filling was done. The automatic algorithm simultaneously segments brain parenchyma, total brain contour, and lesions. As described in the reference [48], the core of algorithm is iterative common mode (ICM) algorithm with classes for background, brain, and MS lesions. We clarified that a more detailed description of the lesion analysis can be found in reference [48].

20. Line 249: What are the exact values for the thresholding of the structural connectivity construction? FOD cut off, number of fibers etc.?

 For the structural connectivity analysis, there is no threshold applied. A probability density map is produced from the tracking methodology, and the mean diffusivities, as described in equation 1, are produced with no thresholding. 

Results

21. Line 335: Table 2 should include the lesion volume information.

 Lesion volume information was added to Table 2.

22. Descriptive statistics that are represented in the Table3&4 are hard to follow in the text. Combining the functional and structural connectivity sections would be better for following the results in the tables that belong to both sections.

 We appreciate the feedback on the structure of the manuscript. To make the tables and text follow more naturally, we elected to categorize the results by analysis. There are now separate sections for “Group differences,” “Impact of time,” and “Imaging and behavioral measures.” Results for all imaging measures are presented in each section, mirroring the presentation in the tables and figures. We hope that this makes the manuscript more readable.

23. Line 417-418: “In the interest of space, only behavioral measures that showed significant relationships are included.” Which relationship? Did you mean that only those that show a significant correlation between at least one of the SFCI components and one of the neuropsychological tests are given in the table?

 The reviewer is correct. We clarified this in the text as follows: “Tables 5 and 6 show Pearson correlation coefficients between imaging and behavioral measures in participants with MS. In the interest of space, only behavioral measures that showed a significant relationship to at least one imaging measure are included in the tables.”

24. Line 431-446: Instead of the terms “positively/negatively”, explaining like “lower performance by SDMT or etc related with the lower/stronger/higher connectivity” may be easier to read.

 This section has been revised to more clearly state the relationships between the variables.

Discussion

25. Line 448-449: “This work demonstrates that a combined structural and functional connectivity metric can be a sensitive tool for identification of neurological disease” there is no other type of neurological disease that shows different patterns etc than MS or identify MS. So this sentence may be confusing. Rather it can be “This work demonstrates that a combined structural and functional connectivity metric can be a sensitive tool for identification of clinical impairment in neurological disease such as MS.”

 This sentence has been modified as suggested by the reviewer.

26. Methodological limitations must be discussed.

 A paragraph discussing methodological limitations has been added near the end of the discussion section.

27. Line 497-500: I respectfully disagree with the authors that normalization is not necessary to compensate bias field effect (i.e noise) either it is region based or not. Before the calculation of the ROIs, registration to a standard atlas is mandatory which the current study also involved registration. This issue should be discussed more. Perhaps, exemplify the order of the analysis for this study compared to other studies.

 With all due respect to the reviewer, the authors stand by the statement that the methodologies employed here don’t require registration to an atlas or common space. All ROI’s and resultant pathways are, in the final determination, determined within subject and in the subject’s native space. For the PCC/AMTL ROI’s, an anatomic landmark is used as a starting point. However, the final ROI, which is very small in relation to the starting region, is determined functionally, within subject in native space. We have clarified that in the paragraph referenced.

Reviewer #2: 

1. line 150 : 256 x 128 might be a matrix error? ( siemens has this 50% distance factor - or gap- of the voxel on MPRAGE sequences in the direction of acquisition , but this does not change the matrix). I suppose also axial acquisition has to be verified, because 176 slices is typical of sagittal acquisitions.

 The authors apologize that it was not made clear that this was an axial 3D MPRAGE. The text has been modified to reflect that. The acquisition matrix was 256 in frequency, 128 in phase and 176 in the slice direction. It was a single slab, 3D acquisition, so there is no distance factor issue. The images were reconstructed to 256x256, but the acquired matrix and slices were as stated in the manuscript.

2. liines 148-171 : please keep same order in the parameter description of each sequence (e.g. slice number,, thickness, FOV, matrix, voxel size etc). Also please report the acquisition duration, confirm that this is the order of acquisition and describe experimental setting , paradigm , pre-scan instructions and training. 

 The scan descriptions have been made more uniform, acquisition durations have been added to each scan, and the order of acquisition has been confirmed. Two paragraphs were added prior to the scan descriptions that include information on the tapping paradigm and participant training. 

3. line 172 MRI data analysis:: please provide all tools that you use in every step i.e. to apply Gaussian filter, fitting the boxcar model, the LV and on. I suppose that this can be provided as supplementary info. Also more info is needed for the experimental set, e.g. bite bar, physiological parameter acquisition. This is to ameliorate reproducibility.

 We added more information on tools throughout the “MRI data processing” section. We also added more information on the finger tapping task and bite bar under “MRI acquisition.” Finally, additional information on physiological monitoring was added under “MRI data processing.”

4. line 219-220: How did you manually modify ROIs, e.g. by erosion of the voxels found in CSF?

 This sentence has been modified to clarify the procedure for ROI modification: “ROIs were visually inspected and, if necessary, manually eroded to ensure all voxels were located in gray matter and not in intragyral CSF (Fig. 2a, lower right).”

5. line 230-242 : please add details for registration and transformation of images and refer to different spaces in a more explicit way: individual space can be functional , DTI, T13D or FLAIR. What kind of interpolation did you use to apply a transformation matrix to a flat ROI?

 Additional details of image registration and transformation have been added to the first paragraph under “Seed region definition.” We used nearest neighbor interpolation for the ROI transforms, and this information is now included. References to specific image spaces have been clarified. 

6. Fig3 : the 3D glass brain image appears blurred and does not permit a good evaluation of the localization.

 Figure 3 was redone to provide a sharper image and allow clearer localization. 

7. line 265: by "cluster size 60" do you mean cluster size threshold of 60 voxels? please rephrase.

 This was rephrased to state ”(p < 1.0×10-4, cluster size threshold = 60 voxels).”

8. line 475-476: some grammar error, please rephrase

 This sentence has been rephrased as: “A possible explanation is that the prior study used a DTI acquisition that had higher spatial resolution and was tailored to target the PCC-AMTL pathway.”

9. line 479 : rather than finding it is a missed relationship or no finding.

 This sentence has been revised to say “…future work is required to clarify the lack of relationship seen here.”

10. It would be desired the availability, when published, of anonymized metadata of the group analysis, for example z-score maps and demographic, clinical and the other metrics that have been used, upon which to make a power analysis by anyone interested: it would be an important addition for the usefullness of the paper and would fulfill the journal's policy and mission.

 The Cleveland Clinic Institutional Review board has strict requirements for sharing individual-level data, particularly demographic and clinical information. We are currently seeking approval from the Cleveland Clinic IRB to share a de-identified dataset. If accepted, the authors agree to share group analysis results and other metadata, as permitted by our institution and within ethical and legal guidelines. We propose to share these data through figshare.

---

## [Decision Letter · Decision Letter 1]

10 Mar 2021

PONE-D-20-32068R1

Evaluation of a connectivity-based imaging metric that reflects functional decline in Multiple Sclerosis

PLOS ONE

Dear Dr. Koenig,

Thank you for submitting your manuscript to PLOS ONE. After careful consideration, we feel that it has merit but does not fully meet PLOS ONE’s publication criteria as it currently stands. Therefore, we invite you to submit a revised version of the manuscript that addresses the points raised during the review process.

As you will see below, most of the remaining comments raised from Reviewer are matters of preference. As such, I leave you some leeway in deciding which to specifically address. However, please justify your responses, accordingly. I do recommend, though, that you update some of the references, as suggested. 

We look forward to receiving your revised manuscript.

Kind regards,

Niels Bergsland

Academic Editor

PLOS ONE

Journal Requirements:

Reviewers' comments:

Reviewer's Responses to Questions

**Comments to the Author**

1. If the authors have adequately addressed your comments raised in a previous round of review and you feel that this manuscript is now acceptable for publication, you may indicate that here to bypass the “Comments to the Author” section, enter your conflict of interest statement in the “Confidential to Editor” section, and submit your "Accept" recommendation.

Reviewer #1: All comments have been addressed

Reviewer #2: All comments have been addressed

2. Is the manuscript technically sound, and do the data support the conclusions?

Reviewer #1: Yes

Reviewer #2: Yes

3. Has the statistical analysis been performed appropriately and rigorously? 

Reviewer #1: Yes

Reviewer #2: Yes

4. Have the authors made all data underlying the findings in their manuscript fully available?

Reviewer #1: Yes

Reviewer #2: No

5. Is the manuscript presented in an intelligible fashion and written in standard English?

Reviewer #1: Yes

Reviewer #2: Yes

6. Review Comments to the Author

Reviewer #1: Overall the review was performed well, the paper has certainly improved. Some remaining points:

Although the authors improved readability of the paper a lot, the references are too old. Cognition in MS is highly topic in recent years and I believe there are more updated relevant works in the literature.

Abstract:

- 3 digits after decimal point is enough for the p value description and 2 digits for r values.

Materials and methods:

- Line 176: Please rephrased as “176 axial slices and 0.94 mm slice thickness”

-“Acquisition time (TA)” would be better instead of “Scan length”

-Please provide abbreviations such as inversion time (TI), TE, TR ….

Results:

In general each table and figure caption should include the abbreviations.

-Figure 4: Please provide all abbreviations MS, SMC… and descriptions like (A) shows sensorimotor cortex etc. in the figure caption.

Reviewer #2: All my questions have been successfully addressed. The only concern of mine remaining is that full data availability has not been definitely addressed

7. PLOS authors have the option to publish the peer review history of their article (what does this mean?). If published, this will include your full peer review and any attached files.

Reviewer #1: **Yes: **Arzu Ceylan Has Silemek

Reviewer #2: **Yes: **Nikolaos Petsas

---

## [Author Response · Author response to Decision Letter 1]

22 Apr 2021

Thank you again to the reviewers for the time and effort applied to our manuscript. Please find our responses to your comments below.

Reviewer #1:

1. Although the authors improved readability of the paper a lot, the references are too old. Cognition in MS is highly topic in recent years and I believe there are more updated relevant works in the literature.

 Updated references have been added. Please note that using the “track changes” function to note reference changes made the manuscript very difficult to read. Rather, the new references are in red type.

Abstract

2. 3 digits after decimal point is enough for the p value description and 2 digits for r values.

 This was modified.

Materials and methods

3. Line 176: Please rephrased as “176 axial slices and 0.94 mm slice thickness”

 This was modified for each scan description.

4. “Acquisition time (TA)” would be better instead of “Scan length”

 This was modified for each scan description.

5. Please provide abbreviations such as inversion time (TI), TE, TR ….

 These definitions were added under Scan 1. 

Results

6. In general each table and figure caption should include the abbreviations.

 Abbreviations have been added to the figures and tables. For the sake of space, a number of captions refer the reader to abbreviations under Table 3. 

7. Figure 4: Please provide all abbreviations MS, SMC… and descriptions like (A) shows sensorimotor cortex etc. in the figure caption.

 Abbreviations and additional description have been added for this figure.

Reviewer #2: 

1. The only concern of mine remaining is that full data availability has not been definitely addressed.

 As previously mentioned, the Cleveland Clinic has strict requirements for data sharing, including review by the Institutional Review Board, legal department, and cybersecurity. We hoped to receive full approval by the resubmission deadline, but still await approval from the Cleveland Clinic legal department. We expect full approval by the end of April 2021. We assure the reviewers and editors that we have moved as quickly as possible to gain approval, and apologize for the delay. The data supporting this work is uploaded to Figshare. As soon as approval is received, the data will be published publically and we will notify the Editor. In the meantime, we have made a private link available to the reviewers and editors: 

https://figshare.com/s/db2f3272926c015412fb

---

## [Editor Report · Decision Letter 2]

26 Apr 2021

Evaluation of a connectivity-based imaging metric that reflects functional decline in Multiple Sclerosis

PONE-D-20-32068R2

Dear Dr. Koenig,

We’re pleased to inform you that your manuscript has been judged scientifically suitable for publication and will be formally accepted for publication once it meets all outstanding technical requirements.

Kind regards,

Niels Bergsland

Academic Editor

PLOS ONE
---

## [Editor Report · Acceptance letter]

31 May 2021

PONE-D-20-32068R2 

Evaluation of a connectivity-based imaging metric that reflects functional decline in Multiple Sclerosis 

Dear Dr. Koenig:

I'm pleased to inform you that your manuscript has been deemed suitable for publication in PLOS ONE. Congratulations! Your manuscript is now with our production department. 

Kind regards, 

on behalf of

Dr. Niels Bergsland 

Academic Editor

PLOS ONE